# Research on Microgrid Optimal Dispatching Based on a Multi-Strategy Optimization of Slime Mould Algorithm

**DOI:** 10.3390/biomimetics9030138

**Published:** 2024-02-23

**Authors:** Yi Zhang, Yangkun Zhou

**Affiliations:** College of Electrical and Computer Science, Jilin Jianzhu University, Changchun 130000, China; yangkunzhou@hotmail.com

**Keywords:** microgrid, optimize scheduling, renewable energy, slime mould algorithm, swarm intelligence

## Abstract

In order to cope with the problems of energy shortage and environmental pollution, carbon emissions need to be reduced and so the structure of the power grid is constantly being optimized. Traditional centralized power networks are not as capable of controlling and distributing non-renewable energy as distributed power grids. Therefore, the optimal dispatch of microgrids faces increasing challenges. This paper proposes a multi-strategy fusion slime mould algorithm (MFSMA) to tackle the microgrid optimal dispatching problem. Traditional swarm intelligence algorithms suffer from slow convergence, low efficiency, and the risk of falling into local optima. The MFSMA employs reverse learning to enlarge the search space and avoid local optima to overcome these challenges. Furthermore, adaptive parameters ensure a thorough search during the algorithm iterations. The focus is on exploring the solution space in the early stages of the algorithm, while convergence is accelerated during the later stages to ensure efficiency and accuracy. The salp swarm algorithm’s search mode is also incorporated to expedite convergence. MFSMA and other algorithms are compared on the benchmark functions, and the test showed that the effect of MFSMA is better. Simulation results demonstrate the superior performance of the MFSMA for function optimization, particularly in solving the 24 h microgrid optimal scheduling problem. This problem considers multiple energy sources such as wind turbines, photovoltaics, and energy storage. A microgrid model based on the MFSMA is established in this paper. Simulation of the proposed algorithm reveals its ability to enhance energy utilization efficiency, reduce total network costs, and minimize environmental pollution. The contributions of this paper are as follows: (1) A comprehensive microgrid dispatch model is proposed. (2) Environmental costs, operation and maintenance costs are taken into consideration. (3) Two modes of grid-tied operation and island operation are considered. (4) This paper uses a multi-strategy optimized slime mould algorithm to optimize scheduling, and the algorithm has excellent results.

## 1. Introduction

Energy shortages become more and more severe with the rapid increase in electricity usage. Utilizing renewable energy sources and optimizing the scheduling of energy structures can effectively solve these problems [1]. Traditional power generation methods have become controversial, such as thermal power plants. Using non-renewable energy sources like coal exacerbates resource depletion and environmental pollution. In contrast, pure low energy utilization of thermal power generation and power generation efficiency lead to high costs [2]. Microgrids offer flexibility, safety, and dispatch ability as a distributed power generation mode. They can alleviate the pressure on the primary grid, supply power, and operate in grid-connected and island modes [3]. When used for power generation, wind, and solar energy are prevalent clean and renewable sources that reduce environmental pollution and resource loss. Microgrids incorporating wind power generation are distributed systems designed to address these issues, aligning with green development, and providing a secure power supply. Optimizing the scheduling of microgrids is an area of growing interest, as it can minimize power generation costs while meeting regional load requirements. However, combining wind, solar, and electricity scheduling involves a high-dimensional 24 h planning problem. Considering various microgrid constraints, developing an efficient method to solve the scheduling problem presents a significant challenge [4].

Most scholars used traditional mathematical models to construct this problem. The short-term optimal scheduling of microgrids can be attributed to 0–1 programming problems, linear programming problems [5], or dynamic programming problems [6]. Mathematical methods were employed to solve them. Ping L et al. used the Lagrange relaxation method to solve the distributed state of the AC-DC hybrid microgrid [7], B Knueven et al. used the mixed integer programming model to control the power generation of the unit [8], and C Ning et al. used a Bayesian non-parametric approach to solve the data-driven dispatch problem under uncertainty in wind power generation [9]. Many mathematical methods solve only the problem but could be more efficient, especially when facing large-scale, multi-objective, and multi-modal problems. Mathematical methods require significant time to calculate and cannot guarantee calculation accuracy and efficiency when dealing with many constraints, such as in microgrids.

In recent years, meta-heuristic algorithms have rapidly advanced and solved many practical engineering problems [10]. These algorithms are advantageous in addressing single-objective, multi-objective, continuous, and discrete problems, ensuring high computational efficiency and accuracy when tackling complex challenges. They exhibit remarkable flexibility and applicability, effectively avoiding local optimal solutions compared to traditional mathematical methods and have been widely applied to various scientific problems [11]. As an essential branch of meta-heuristic algorithms, swarm intelligence algorithms are commonly employed in continuous and discrete problems within practical engineering contexts. Microgrid economic optimal dispatching has increasingly been addressed using swarm intelligence algorithms, as it is a typical high-dimensional, persistent optimization problem. Cui Z et al. solve mathematical problems using a hybrid cuckoo algorithm [12]. Microgrid dispatch considering cogeneration also uses a swarm intelligence algorithm [13]. The swarm intelligence algorithm to solve this problem has become mainstream according to the in-depth research on microgrid scheduling. Abhilipsa et al. [14] solved the problem of randomness in non-renewable energy and proposed a bidding strategy to optimize the profit function, but the constraints in the proposed model cannot be well integrated with the algorithm. Dai et al. [15] used simulated annealing algorithm and particle swarm algorithm for fusion, which enriched the diversity of individuals searched. However, the optimization algorithm proposed was highly random in search and not efficient. Cao et al. [16] studied microgrid dispatching strategies in two modes: grid-connected operation and networked operation. However, the paper did not intuitively display the power distribution diagram of each unit. Wang et al. [17] used a joint dispatch to solve the optimal dispatch problem of microgrids, but joint dispatch in multiple areas sacrifices the benefits of a single model, and the whale algorithm cannot balance this process well. Zhang Y et al. use the butterfly algorithm to optimize microgrids [18]. Wu, Zhi et al. [19] used particle swarm optimization to optimize the CHP microgrid system, which can meet the scheduling requirements, but the search accuracy of the particle swarm algorithm needs to be higher, and it cannot guarantee a good solution effect. Marzband, Mousa et al. [20] used the ant colony algorithm to optimize the scheduling of the microgrid. The experimental results show that the ant colony algorithm is worse than the particle swarm optimization algorithm for this problem. However, the performance of the ant colony algorithm could be better than the discrete problems in solving numerical problems. Yeh, Wei-Chang et al. [21] proposed a new concept involving the regeneration of a new feasible variable set and used a new genetic algorithm for optimization without penalty coefficients. The accuracy of the algorithm still needs to be improved. This paper proposed a novel algorithm to solve this practical application problem. After the slime mould algorithm was created, it was successively used in high-latitude, multi-objective numerical problems. Similarly, the microgrid model simulated in this paper is a complex numerical problem, and related SMA articles inspired us. The slime mould algorithm (SMA) is a novel swarm intelligence algorithm proposed by Li in 2020 [22]. Researchers observed that slime mould colonies stretch or expand their vein-like tubes during foraging to control the concentration of biomass flow. The SMA was developed based on this phenomenon. The SMA effectively balances exploration and exploitation performance compared to other swarm intelligence algorithms. Although the SMA has demonstrated excellent results in solving continuous problems [23], it has drawbacks, such as slow convergence speed and susceptibility to local optima. Liu et al. [24] proposed a decentralized transaction model based on the master-slave game, using the slime mould algorithm to optimize the model, reducing energy storage pressure and improving efficiency, but they did not consider the cost of environmental factors. Emad A. Mohamed et al. [25] used an optimized fractional-order controller based on the slime mould optimization algorithm (SMA) to establish an improved coordination method and used it for optimal scheduling of microgrids. However, the optimization of SMA in the article was more limited to adaptive weights. Other strategies are ignored. Pawan Kumar Kushwaha et al. [26] proposed a techno-economic environment (TEE) design of off-grid microgrid (OGM) using SMA to improve rural power reliability. Compared with PSO, SMA shows better performance, but the article does not study the grid-connected operation.

Considering that although the above research can basically complete the optimal dispatch of microgrids, there are problems such as low accuracy, simple optimization goals, and few types of distributed power sources. This paper studies two modes of microgrids, grid-connected operation and islanded operation, taking into account the environment costs, operating costs, maintenance costs and other costs needed to fully meet the reliability, and economic and environmental protection of the microgrid.

This paper introduces an improved SMA to solve the microgrid optimal scheduling problem: the multi-strategy fusion slime mould algorithm (MFSMA). This paper adds a variety of strategies to the SMA because the search accuracy is not high. And the search strategy is also added to the SMA because the follower operator can improve the convergence speed of the solution, which together form the MFSMA. By employing reverse learning to expand the search space, MFSMA prevents falling into local optima. Furthermore, new adaptive parameters ensure dynamic and adaptive search during iterations. The search pattern of the salp swarm algorithm is also incorporated to accelerate convergence. The simulation results show that when solving high-dimensional, multi-objective problems such as microgrid optimal dispatch, MFSMA has faster convergence speed, higher search accuracy and stronger robustness than other algorithms, and it achieves a reduction in the total cost of the microgrid.

The contributions of this paper are as follows: (1) A comprehensive microgrid dispatch model is proposed, which includes diesel generator (DG), wind turbine (WT), photovoltaic (PV), fuel cell (FC) and micro turbine (MT) as well as load shedding. (2) Environmental costs, operation and maintenance costs are taken into consideration, and the economic and environmental protection of the microgrid are achieved at the same time. (3) Two modes of grid-tied operation and island operation are considered. (4) This paper uses a multi-strategy optimized slime mould algorithm to optimize scheduling, and the algorithm has excellent results.

The paper is structured as follows: Section 2 models the optimal economic dispatch for the microgrid. Section 3 presents the basic SMA algorithm, details the optimization improvements of MFSMA for the SMA method, and provides the flowchart for solving the microgrid optimal dispatching problem. Section 4 benchmarks MFSMA against other algorithms. Section 5 applies MFSMA to the microgrid optimal dispatching problem and compares convergence speeds with different algorithms. Finally, Section 6 summarizes and provides an outlook.

## 2. Problem Formulation and Microgrid Model

The optimal scheduling model of the microgrid is divided into multiple parts to supply power to the microgrid, including the operation of one or more micro turbines (MT), fuel cells (FC), diesel generators (DG), wind power (WT), photovoltaic (PV), and battery as shown in Figure 1. The microgrid has two operating modes, grid-connected operation and islanded operation. Grid-connected operation can purchase power from the main grid, while islanded operation is a relatively closed operation mode. This paper proposes a 24 h optimal dispatch model, in which distributed generating units are combined to generate electricity according to load demand [27]. The power generation process will generate operating costs and maintenance costs. In addition to wind power generation and photovoltaic power generation, harmful gases will also be produced which will cause environmental pollution, so there are also environmental costs. The research content of this article is to reduce the total cost as much as possible while satisfying the constraints. The cost formulas for various units are introduced below.

### 2.1. Diesel Generator Model

The mathematical model of the diesel generator is similar to the mathematical model of coal power generation in thermal power plants, and its power consumption curve is a complex mathematical problem [28]. The unary quadratic function of Formula (1) calculates the consumption cost of the diesel generator. Among them, where $a_{i}$, $b_{i}$, $c_{i}$ are the fuel cost coefficients which are used to adjust the relationship between fuel consumption and power, $e_{i}$ and $f_{i}$ are coefficients that are related to valve points of the ith generation unit. $P_{i,t}$ represents the average power of unit i at time t. The power is directly proportional to the cost. $C_{DG}$ represents the energy consumption formula of the diesel generator,n represents the number of diesel generator.
(1)$$C_{DG,t}=a_{i}+b_{i}\cdot {P_{i,t}}^{2}+c_{i}\cdot P_{i,t}^{3}+\left|d_{i}\cdot \sin\left(e_{i}\cdot \left(P_{i,min}-P_{i,t}\right)\right)\right|$$

### 2.2. Wind Power Generation Model

The wind turbine (WT) converts the wind’s kinetic energy through the fan’s rotation into mechanical energy. Then, the wind energy generator starts to work under the fan’s drive and restores the fan’s mechanical energy into magnetic energy, converting magnetic energy into electrical energy [29]. The general conclusion concludes that wind speed is directly proportional to power generation. Then, the output characteristics of wind power generation are shown in Formula (2) when the wind speed is known.
(2)$$P_{WT}=\left\{\begin{array}{c} \begin{array}{cc} 0,\;v<v_{ci}\\ \frac{P_{t}\left(v-v_{ci}\right)}{v_{\mu }-v_{ci}} & v_{ci}<v<v_{\mu }\\ P_{t},v_{\mu }<v<v_{co}\\ 0,v>v_{co} \end{array} \end{array}\right.$$

In the formula, $P_{t}$ is the rated power, $v_{\mu }\;$ represents the rated wind speed, $v_{ci}$ represents the rated wind speed, $v_{co}$ is the cut-out wind speed, and $P_{WT}$ is the output rated power. Wind power is clean energy whereby its environmental pollution is negligible. The proportion of wind power generation should be maximized as much as possible in the microgrid economic dispatch. The output power of wind turbines can be obtained by predicting the wind speed at a particular moment [30] simultaneously.

### 2.3. Solar Power Model

The output characteristics of the power generated by solar cells are related to their volt-ampere characteristics. The output power of solar cells fluctuates significantly, and many factors affect the conversion efficiency of solar cells, such as solar radiation, temperature, weather conditions, battery internal resistance, material characteristic factors, etc. Solar radiation intensity, temperature, and load characteristics are the most important influencing factors [31]. The output power of photovoltaic power generation can be calculated by the following Formula (3).
(3)$$P_{pv}=P_{STC}\cdot \frac{G_{ING}}{G_{STC}}\cdot \left[1+k\cdot \left(T_{c}-T_{r}\right)\right]\;$$

In this formula, $P_{STC}$ represents the maximum output power of the solar generator set under standard conditions, $G_{STC}$ represents the solar radiation intensity under the standard requirements which generally take $G_{STC}=1000$ W/m^2^, and $G_{ING}$ represents the current actual irradiation intensity, k is a characteristic parameter, and $T_{c},\;T_{r}$ are the temperature and reference temperature of the solar cell, respectively. The output power of PV is proportional to the solar radiation intensity and changes with sunrise and sunset. The output power of PV is low in the early morning and night, and the output power of PV is high [32] at noon. At present, there are papers that use machine learning to predict the uncertainty of photovoltaics to ensure a balance between cost and system stability [33]. Suchismita Patel et al. used battery storage in renewable energy systems to reduce power fluctuations caused by the intermittent behaviour of renewable energy sources and proposed that a controller is applied to a voltage-controlled loop, which is important for the proposed renewable energy system. Energy stabilization strategies are proven [34]. This is also our future research direction and research content.

### 2.4. Storage Battery Model

It is necessary to add batteries to the microgrid unit to cope with the influence of wind speed, solar radiation, temperature, and other uncertain factors on the stability of the grid in the microgrid. The use of batteries can improve the safety and stability of the entire microgrid and play the role of peak shaving and valley filling facing the shortcomings of the fluctuation and randomness of the output power of the microgrid [35]. The battery is symbolically attributed to an energy storage unit for research. Its role is to store the excess electricity generated by the microgrid and quickly provide electricity when the microgrid is insufficient [36]. This paper mainly studies the mathematical model of the storage devices. The cost of the storage battery device ($C_{SB}$) is shown in Equation (4).
(4)$$C_{SB,t}=\alpha \cdot P_{SB,t}+\beta \cdot \;P_{SB,t}+\gamma \cdot P_{SB,t}$$

In the formula, $P_{SB}$ represents the power of the energy storage battery. When $P_{SB}$ is less than zero, it represents charging, and when $p_{SB}$ is greater than zero, it represents discharging. Excess electrical energy can be stored in an energy storage device so that after the load demand is met, it can be discharged to meet the load demand when the power supply is insufficient. The charging and discharging process will consume costs. α,β,γ represents the maintenance coefficient, depreciation coefficient and pollution coefficient of the battery, respectively. This paper will not consider the pollution of wind power generation, in other words, γ=0. There are upper and lower power limits for the energy storage battery. After completing daily scheduling, the energy storage battery is supposed to return to its original state, otherwise it will be punished accordingly.

### 2.5. Micro Turbine Model

Micro turbines are power generation equipment that use natural gas as fuel. Their capacity is generally small. The use of a micro turbine causes little pollution and low operation and maintenance costs. It can shut down and startup quickly and has easy installation. It is an ideal choice when it comes to the load of carbon neutrality [37]. This power generation equipment that coexists with economic and environmental protection. Its formula is as follows:(5)$$C_{MT}=p_{gas}\underset{t}{\sum }\frac{P_{MT,t}}{\eta_{MT,t}},\forall t\in T$$

In the formula, $p_{gas}$ is the price of natural gas, $P_{MT,t}$ is the power of the micro turbine at time t, and $\eta_{t}$ is the efficiency at time t.

### 2.6. Fuel Cell Model

Fuel cells, also known as electrochemical generators, are the fourth power generation technology after hydropower, thermal power and atomic power. From the perspective of energy conservation and ecological environment protection, fuel cells are the most promising power generation technology [38]. The fuel cell studied in this paper uses natural gas, and its cost formula is as follows:(6)$$C_{FC}=p_{gas}\underset{t}{\sum }\frac{P_{FC,t}}{\eta_{FC,t}},\forall t\in T\;$$

Similar to Equation (5), the cost of fuel cells is closely related to the price of natural gas, working efficiency, and working power.

### 2.7. Main Grid

In order to face the impact of power failure or severe weather in the microgrid, this paper also studies the grid-connected operation of the microgrid. The microgrid can purchase electricity from the main grid to cope with emergencies or use the price difference to obtain profits. Generally speaking, the prize of the power grid is not fixed every day but is divided into multiple intervals according to the amount of electricity consumption, and is generally divided into valley areas, base areas and peak areas. When the user’s electricity consumption is high, the electricity price of the main network will increase, and vice versa.

### 2.8. Objective

This paper adds multiple cost functions and constraints to the model to ensure the validity of the simulation. The cost of generating electricity from the generator units in the microgrid, the cost of purchasing electricity from the grid, the environmental cost and penalty cost caused by pollutants are considered. The cost of the model studied in this article will be divided into five parts, as detailed below:(1)Controllable energy cost:

The energy that can be regulated in this paper include micro turbines, fuel cells, diesel generators, and energy storage batteries. Their power in each time period can be scheduled by algorithms. When the generators are running, they generally generate operating costs based on power. The formula is as follows:(7)$$C_{1}=\overset{T}{\underset{t}{\sum }}(C_{MT,t}+C_{SB,t}+C_{FC,t}+C_{DG,t})$$

In the formula, t represents the period of a day. This formula means to accumulate operating cost in all time periods. If it is a day’s cost, T=24. Controllable energy is the main decision variable in this paper, and their size greatly affects the final cost [39,40,41].

(2)Uncontrollable energy cost:

The uncontrollable energy sources in this article are generally wind power generation and photovoltaic power generation. The output power of the wind turbine calculated based on wind speed, light intensity, temperature and other parameters will not change, but operating costs will also be incurred based on the power during operation. Wind and solar power generation does not produce polluting gases. Therefore, the proportion of uncontrollable energy power generation should be increased in the optimal dispatch of microgrids.
(8)$$C_{2}=C_{WT}+C_{PV}$$
(9)$$C_{WT}=\overset{T}{\underset{t}{\sum }}P_{WT,t}\cdot K_{WT}$$
(10)$$C_{PV}=\overset{T}{\underset{t}{\sum }}P_{PV,t}\cdot K_{PV}$$

$K_{WT}$ and $K_{PV}$ in the formula is the cost coefficient of wind power and photovoltaics. It is generally a fixed value and is used to comprehensively measure the maintenance costs, depreciation costs and other costs incurred by wind and photovoltaic units during operation.

(3)Power purchase cost:

If the microgrid is connected to the main grid, it can purchase electricity from the grid. The cost of purchasing electricity is directly proportional to the price of electricity. The price of electricity in different periods of the day is different; it is generally divided into peaks, flat peaks, and troughs. The following formula reflects the cost of purchasing electricity from the grid for the microgrid.
(11)$$C_{3}=\overset{T}{\underset{t}{\sum }}\left(C_{grid,t},P_{grid,t}\right)\;\;,\;C_{grid,t}=\left\{\begin{array}{c} C_{Valley}\\ C_{Base}\\ C_{Peak} \end{array}\right.\propto t$$

$P_{grid,t}$ is the power transferred from the grid at time t. $C_{grid,t}$ represents the real-time electricity price.

(4)Environmental cost:


(12)
$$C_{4}=\overset{T}{\underset{t}{\sum }}\left(\overset{M}{\underset{k}{\sum }}C_{k}\left(\overset{N}{\underset{i}{\sum }}r_{ik}P_{i}\left(t\right)+r_{grid}\cdot CGP\left(t\right)\right)\right)$$


The environmental cost in the microgrid economic dispatch model generally refers to the cost of ecological governance after the microgrid emits pollutants. Microgrids result in the need to spend expenses to clean up the environment or pay fines. In general, polluting gases released by microgrids include but are not limited to carbon dioxide, sulfur dioxide, nitrogen oxides, and carbon monoxide [42]. The above formula expresses this cost, where $r_{ik}$ represents the penalty fee rate of a certain pollutant, $C_{k}$ represents the cost per kg of gas treatment, and the detailed penalty table will be given in front of the simulation. $r_{grid}$ represents the electricity purchased from the power grid, and $CGP\left(t\right)$ represents the charging standards for environmental governance of the power grid in different time periods. In this paper N represents N types of power generation methods, and only controllable energy units will produce polluting gases. The first summation represents the sum of all the pollutants produced by all types of power generation in the t period. The second summation means that the costs of different pollutants will be calculated separately and summed, and M represents the number of pollutants, then the third summation means adding up the pollution costs for all time periods.

(5)Startup and shut down cost:

If the power supply is not tight, micro turbines and fuel cells can be shut down, but when it is started again, there will be corresponding ramp-up costs:(13)$$C_{5}=\overset{T}{\underset{t}{\sum }}{C_{MT}}^{\prime }\cdot \underset{\;}{\max}(0,U_{MT,t}-U_{MT,t-1})+{C_{FC}}^{\prime }\cdot \underset{\;}{\max}(0,U_{FC,t}-U_{FC,t-1})$$

Generally speaking, if the status of the generator is running at time t, it is regarded as 1; if the status of the generator is stopped, it is recorded as 0. In the formula, ${C_{MT}}^{\prime }$ and ${C_{FC}}^{\prime }$ are the start-stop cost coefficients of MT and FC, respectively.

From the above analysis, we can see that the objective function of our microgrid economic dispatch model is as follows:(14)$$F=w_{1}\cdot (C_{1}+C_{2}+C_{3}+C_{5})+w_{2}\cdot C_{4}$$

Among them, $w_{1}$ and $w_{2}$ are adjustment coefficients, which are used to adjust the operating cost and environmental benefits of the microgrid. Changing the two weights can change the microgrid’s operation strategy. In this paper, both parameters are taken as 0.5. F is the total cost, The fitness value in simulate is the minimum value of F.

### 2.9. Restrictions

(1)Power balance constraints


(15)
$$P_{d,t}=\overset{N}{\underset{i=1}{\sum }}P_{G,t}+a\cdot P_{gird,t}\;,a=\left\{\begin{array}{c} 1,grid-tied\\ 0,island \end{array}\right.\;$$


In the formula, $P_{d,t}$ represents the system load power at time t, $P_{G,t}$ represents the active power delivered by the microgrid system during t period, and $P_{gird,t}$ represents the microgrid system during t period active power purchased from the grid. N represents all power generation methods, which summed up as the total power generation of the microgrid.

(2)Output power constraints of each generator set


(16)
$$P_{i}^{min}<P_{i}<P_{i}^{max}$$


$P_{i}^{min}$ and $P_{i}^{max}$ in Formula (10) represent the minimum power and maximum power of unit i, respectively, which means that all units must work within reasonable upper and lower limit power.

(3)Exchange power constraints of microgrid and main grid


(17)
$$P_{grid}^{min}<P_{grid}<P_{grid}^{max}$$


Formula (11) indicates that the power exchange between the microgrid and the main grid also has an upper and lower limit, and the system must operate within this range.

(4)Constraints of storage battery units


(18)
$$P_{f,t}^{min}<P_{f,t}<P_{f,t}^{max}$$



(19)
$$U_{SB}^{initial}=U_{SB}^{end}$$


Equation (12) represents the maximum power of charging and discharging the energy storage device. $U_{SB}$ represents the capacity of the energy storage battery. In the simulation of this paper, after completing one day’s dispatch, the energy storage battery should return to its original state.

To sum up, these constraints must be strictly obeyed in the economic optimal dispatching model of the microgrid. If these conditions are violated, the microgrid model will be meaningless [43]. The objective function we set up considers the microgrid’s operating cost and environmental benefits simultaneously and makes the two mutually restrictive to solve the most reasonable microgrid operation strategy.

## 3. Algorithms Improvement

### 3.1. Standard SMA Algorithm

The unique physiological characteristics of slime moulds cause them to produce an oscillating wave during the foraging phase. Moreover, the oscillating wave in the vein forms positive feedback from the biological oscillator in the slime moulds. The oscillating wave produced by the biological oscillator was stronger as the food concentration increased. It causes the slime mould’s venous pipeline to widen and the cytoplasmic flow in the cell to increase. Specifically, slime moulds will use the restricted area search method [44] and focus the search on the food source that has been searched when the food quality is high. If the concentration of the initially found food source is low, the slime moulds will leave this food source in search of a better food source [45]. It demonstrates the adaptive characteristics of the slime mould network in the foraging stage. Taking advantage of this feature, we can use the slime mould algorithm to solve the maximum value of the function. The unique characteristics of the slime mould community can not only make full use of the solution space, but also quickly find the extreme value, so this paper mainly studies the slime mould algorithm. Mathematical modelling of the slime mould system is carried out below:(20)$$X\left(t+1\right)=\left\{\begin{array}{c} X_{b}\left(t\right)+v_{b}\cdot \left(W\cdot X_{A}\left(t\right)-X_{B}\left(t\right)\right),r<p\\ v_{c}\cdot X\left(t\right),r\geq p \end{array}\right.$$

In the formula, *t* represents the current iteration number, *m* represents the number of slime moulds, $X_{b}\left(t\right)$ represents the optimal solution under the current iteration number, $X_{A}\left(t\right)$ and $X_{B}\left(t\right)$ are the two random solutions under the current iteration number, and $X\left(t\right)$ represents the current solution. $r\in rand\left[0,\;1\right]$. $W,v_{b},v_{c},p$ are four important parameters explained below.
(21)$$p=\tanh\left|S\left(i\right)-DF\right|$$
(22)$$a=\tanh^{-1}\left(-\left(\frac{t}{T}+1\right)\right),v_{b}\in \left[-a,a\right]\;$$
(23)$$W\left(SI\left(i\right)\right)=\left\{\begin{array}{c} 1+r\cdot \log\left(\frac{bF-S\left(i\right)}{bF-wf}+1\right),\;\;\;\;i<\frac{m}{2}\\ 1-r\cdot \log\left(\frac{bF-s\left(i\right)}{bF-wf}+1\right),\;\;\;\;i\geq \frac{m}{2} \end{array}\right.$$
(24)$$SI\left(i\right)=sort\left(S\right)$$
where, $v_{c}$ is a function that oscillates between [−1, 1] and eventually approaches 0 with increasing iterations. W represents the width of the venous pipeline, $S\left(i\right)$ represents the adaptive value of the current individual solution, T represents the maximum number of iterations, DF,bF,wf respectively represent the optimal adaptive value in the current entire iteration process [46], the optimal adaptive value under the current iteration number and the worst adaptive value in the current iteration process. $SI\left(i\right)$ represents the adaptive ranking of the slime mould population after an iteration. This part mainly simulated the thickening or thinning of the veins of myxomycetes according to the concentration of food. However, the myxomycetes would not move to the high concentration of food, and several individuals in the myxomycetes population would always be separated to find other food sources [47]. In summary, the location update formula of the myxomycetes population is as follows:(25)$$X\left(t+1\right)=\left\{\begin{array}{c} X_{b}\left(t\right)+v_{b}\cdot \left(W\cdot X_{A}\left(t\right)-X_{B}\left(t\right)\right),r<p\\ v_{c}\cdot X\left(t\right),r\geq p\\ rand\cdot \left(UB-LB\right)+LB,rand<z \end{array}\;\;\right.$$
where  UB,LB represents the upper and lower bounds of the solution space, once the random number is less than z, the individual will search for food in random locations.

### 3.2. Standard Salp Swarm Algorithm

The SSA is a swarm intelligence algorithm proposed by Mirjalili et al. in 2017 [48]. The algorithm simulates the social behaviour of salps as they forage for food. Salp populations are generally divided into two groups of functionally distinct roles, leaders, and followers. In the end-to-end chain structure of salps, the leader is used to trace the food source, and the followers closely follow the individual salps in front of them. This paper used the follower search operator of SSA:(26)$$x_{j}^{i}=\frac{1}{2}\left(x_{j}^{i}+x_{j}^{i-1}\right)$$
where i≥2  and $x_{j}^{i}$ shows the position of ith follower salp in jth dimension, this formula describes the process by which each follower approaches the previous individual in a salp population [49].

### 3.3. Multi-Strategy Fused Slime Mould Optimization Algorithm (MFSMA)

MFSMA has both the search characteristics of the SMA which can effectively use the search space and has the advantages of the SSA which can provide a fast search process. They have similar search subgroups. And the search modes can be mutually optimized through coding, thus integrating the advantages of each algorithm. In this subsection, we will propose a multi-strategy fusion slime mould algorithm (MFSMA) for solving the microgrid optimal dispatching problem model. Aiming at the shortcoming of the SMA, some improvement schemes are proposed.

#### 3.3.1. Refracted Opposition-Based Learning

Opposition-Based learning is a strategy for optimizing swarm intelligence algorithms. The idea of this strategy is to generate a reverse solution through the current solution, compare the fitness values of the two, and select the best one for use. The Refraction Opposition-Based Learning strategy (ROBL) is an optimization strategy combined with the principle of light refraction [50]. ROBL can provide more choices for the algorithm and prevent inefficiency.

Usually, in a refraction situation, we can find the refractive index:(27)$$x_{i,j}^{*}=\frac{a_{j}+b_{j}}{2}+\frac{a_{j}+b_{j}}{2\delta }-\frac{x_{i,j}}{\delta }$$

$x_{i,j}$ is the position of the ith slime mould in the jth dimension, $x_{i,j}^{*}$ is the refraction reverse solution of $x_{i,j}$. $a_{j}$, $b_{j}$ are the upper and lower limits of the search space; the refraction solution can be generated by changing δ and n, and can never jump out of the local optimal solution and approach the global optimum.

#### 3.3.2. New Adaptive Parameter

In the original SMA, $v_{c}$ does not have effective adaptability [51]. In the SMA author’s article, $v_{c}$ oscillates between [−1, 1]. We set a new parameter b, its formula is as follows:(28)$$b={\left(e^{\frac{T-t}{T-1}}\cdot \frac{1}{e-1}\right)}^{2}$$

The original $v_{c}$ cannot guarantee sufficient ability for exploration and jumping out of the local optimum. Although there is also an adaptive balancing process, the overall smaller value of $v_{c}$ does have the problem of premature convergence. Our new adaptive parameters *b* and $v_{c}$ can guarantee the solution accuracy and convergence of the algorithm.

#### 3.3.3. Follower Strategy

Aiming at the shortcoming of SMA’s convergence speed, the follower’s movement strategy in SSA is added to SMA. First, after one iteration, we sort the slime mould group according to the fitness value from large to small, and select a part of the population, generally selecting the individuals at the back of the population, and use Equation (26) to optimize the group. The foraging strategy of the followers in the salp group can make up for the shortcomings of the slime mould algorithm and improve the convergence speed and solution accuracy.

#### 3.3.4. MFSMA for Solving Microgrid Optimal Dispatching Problem

In Algorithm 1, we introduce the process of MFSMA to solve the microgrid optimal dispatching problem. First, the power of the load needs to be input, that is, the power demanded by the user, followed by the power of wind power generation and photovoltaic power generation, and finally the upper and lower limits of the power generated by each unit in each period, and the corresponding characteristic coefficient and electricity price. In this paper, all the input data are divided into 24 h to optimize the scheduling of the microgrid. First, in the early stage of the algorithm, a multi-dimensional array is formed in each slime mould individual. These arrays are used to store the power information of each unit. This process is called the initialization of the slime mould community. Then, multiple strategies of the slime mould algorithm are used to iteratively update the community information, and a feasible solution is generated under the premise that the constraints are met. The optimal solution is updated until the maximum number of iterations is reached. This is the entire search process of MFSMA. Figure 2 is the flow chart of MFSMA.
**Algorithm 1****Input:**N: Number of the units (dimension of the model)Pmin,Pmax, P: The upper and lower limits of the output power of each unit, Load power, wind power generation and photovoltaic power generation by time periodk: The characteristic coefficient of each unit**Output:** Minimum total cost of microgrid power generation1: Initialization parameter popsize, Max_iteraition;2: Initialization the position of slime mould Xi,ji=1,2,3,…,popsizej=1,2; 3: Set the iteration counter ***it*** = 0 4: While it < Max_iteraition, then 5:     Calculate the fitness of all slime mould by Equation (14);6:     Update bestFitness, Xb7:     Calculate the W by Equation (23);8:    For each search portion9:       Update p,vb,vc;10:      Update positions by Equation (20);11:      Generate refraction population by Equation (27);12:          if the  fitness of xi,j*>the  fitness of xi,j13:             xi,j= xi,j*;14:          end if 15:    End for16:    Sort X;17:    For ***i*** = 1: ***popsize***18:      if i ≥sortindex/219:       xi,j Updates position by Equation (26);20:      end if21:    end for22:    it = it + 1;23: End while 24: Return bestFitness,Xb;

## 4. Comparison

In this section, we will use other functions to compare with MFSMA. For the numerical algorithm, we use many benchmark functions for comparison. The computer environment is as follows: the operating system is Windows 11, the CPU is Inter i7-13700H, memory is 16 GB. The algorithms for comparison were coded by MATLAB R2021a.

### 4.1. MFSMA Qualitative Analysis

Figure 3 shows the qualitative analysis results of MFSMA in the processing of unimodal and multimodal functions (Sphere, Griewank, Rastrigin, Ackley, Rosenbrock), and visualizes the changes in the position and fitness value of the optimized slime mould during the foraging process. This graph consists of three indicators: search history, slime mould average fitness value, and iteration convergence curve. In this part, we set the maximum number of iterations of MFSMA to 200 and the population size to 50.

The slime mould has carried out a cross-search pattern near the optimal solution by observing the history graph which ensures the accuracy of the search solution. More slime moulds are gathered in the area where the gradient drops faster, which proves that our algorithm has a faster convergence speed. In the multimodal function, a small number of slime moulds are concentrated in the local optimal area, which reflects the choice of the algorithm for the global optimal and local optimal solutions. By observing the population average curve, the algorithm oscillates quickly in the early stage and smaller in the later stage, and the oscillation frequency is inversely proportional to the number of iterations. This reflects the high adaptability of the slime mould in different functions and the rationality of the search, which balances the search space and convergence speed well, and has strong robustness. Finally, it can be seen from the convergence curve that the algorithm converges at a very fast convergence speed, which ensures the efficiency of the algorithm when solving functions.

### 4.2. MFSMA Compared with Other Algorithms

This section uses several other algorithms to compare with MFSMA, and many functions are used for comparison. Other algorithms include slime mould algorithm (SMA), whale optimization algorithm (WOA), grey wolf optimizer (GWO), dung beetle optimizer (DBO) [52], particle swarm optimization (PSO), salp swarm algorithm (SSA), and Jaya algorithm [53]. We set the population size of all algorithms to 30, the maximum number of iterations to 200, and the results of 20 times were averaged to ensure fairness. This paper evaluated the results using standard deviation (STD) and mean (AVG). Table 1 is functions; Table 2 is results. Figure 4 shows the convergence curve of nine algorithms.

This paper uses a variety of unimodal, multimodal, and fixed-dimensional functions to compare the performance of the algorithms. It can be seen from the data in Table 2 that when solving these benchmark functions, MFSMA can basically find the optimal solution. At the same time, the algorithm proposed in this paper has a smaller standard deviation, which shows that MFSMA has higher solution accuracy and robustness.

Figure 4 shows the convergence curves of nine algorithms when facing these benchmark functions. It can be seen from the figure that MFSMA has searched the area near the optimal solution very early, which shows that MFSMA has the highest convergence speed. In each iteration after completion, MFSMA is located at the bottom of all the curves, indicating that the algorithm has the highest accuracy. The comprehensive test results of the benchmark function indicate that the algorithm proposed in this article is relatively excellent.

## 5. Simulation

This paper simulates two aspects of the grid-connected operation and island operation, and studies two power supply modes in two regions. The load and output power of different networks are different, but their pollutant penalty standards are roughly the same.

### 5.1. Grid-Connected Operation

In this section, we will conduct a specific application research, that is, the study of the economic optimal dispatching problem of microgrids. This microgrid contains diesel generators, wind power and photovoltaics, and is able to connect to the grid. First, we used MFSMA to solve this problem, and then compared MFSMA with SMA, SSA, and slime mould algorithm with adaptive differential evolution algorithm (SMA-ADGE) [54], slime mould algorithm- seagull optimization algorithm (SMA-SOA) [55] and other swarm intelligence algorithms to compare the problem-solving effect. The specific parameters are given in the chart below. Table 3 and Table 4 are the cost technology of the diesel generator [56]. $P_{min}$ and $P_{max}$ represents the upper and lower limits of the installed capacity of each unit.

In Table 4 we give the microgrid pollution control costs and fine rate.

Typical 24 h loads and real-time electricity prices ($/kW·h) are given in Figure 4 [57,58]: we plotted the wind power generation power and photovoltaic power generation power for 24 h on a certain day.

This paper simulated the optimal dispatch of different types of DGs in the micro-grid, using the four models given in Table 3, DG1-4 correspond to a-d in the figure respectively. It can be seen from Figure 5 that both wind power generation and photovoltaic power generation are unstable. The output power of wind power generation is mainly affected by the real-time wind speed, while photovoltaic power generation is greatly affected by solar radiation and temperature. Photovoltaic generators produce little electricity during periods of low solar radiation. We use MFSMA to optimize the dispatch of the economic model of the microgrid. This paper uses a variety of electrical constraints to meet the load of 24 h a day. By controlling the active power output of the DGs that can control the operating cost and environmental governance cost of the microgrid, we reduce the total cost and complete the economy of the microgrid.

From Figure 6, MFSMA can be used to optimize the scheduling of the microgrid to meet the demand of the load. We concluded that the general rule is that when the active power output of PV is less, the microgrid is more dependent on the use of wind power generation that is less related to temperature and solar radiation for power output. Secondly, the power output of the turbine unit results in changes in the output power which can effectively reduce production costs. Once the power supply is insufficient, the microgrid will choose to purchase electricity from the grid or use the battery for charging. If the output power is greater than the load, the microgrid will charge the battery for emergency use, demand, or sell to the grid. It can be clearly seen from the figure that our algorithm constantly adjusts the active output power of the gas turbine unit according to different objective conditions. This dynamic adjustment can effectively avoid the peak period of electricity consumption and rationally utilize the trough period. Through this optimizing scheduling, it can keep the total cost of microgrid operation at a low level.

In Figure 7, we use MFSMA, SMA, SSA, SMA-ADGE, and SMA with SOA to apply the economic optimal dispatch model of the microgrid, and compare the total costs obtained by them. DG1-4 in Table 3 correspond to a-d in the figure respectively. All iterations were performed 1000 times and an average of 30 times was taken.

The MFSMA has the fastest convergence speed and the highest solution accuracy compared with other SMA-derived algorithms and the salp swarm algorithm. From the initial stage of the iteration from Figure 7, the number of iterations to reach the optimal solution of MFSMA is 25.17–75.54% for other algorithms, and the quality of MFSMA’s optimal solution is 7.08–28.5% higher than other algorithms. The MFSMA began to accelerate the convergence and maintain the fastest convergence speed, thanks to the expansion of the search space by ROBL, so that the MFSMA can traverse more possible optimal solutions in a short period of time. In the middle and later stages of the algorithm, the MFSMA has a higher convergence speed and solution accuracy, and due to the salp swarm follower operator, the fast cluster search mode can grab better solutions and jump out of local optimal solutions. It is worth mentioning that the adaptive optimization coefficient controls the process from beginning to end, which ensures the overall stability of the algorithm and improves the robustness of the algorithm. When solving the multi-objective optimization microgrid model of multiple units, the number of dimensions to be solved is high, so the requirements for the algorithm are correspondingly increased.

### 5.2. Island Operation

This section studies the situation of island operation. Micro turbines, fuel cells, energy storage batteries and wind and solar power generation are considered in the microgrid. This paper used MFSMA and SMA, WOA, GWO, and DBO for comparison.

The situation of wind power generation and photovoltaic power generation is shown in Figure 8. Other electricity charges, pollutant charges and grid-connected operation are the same. The parameters of micro turbine, fuel cell and energy storage battery are shown in Table 5.

It can be seen from Figure 9 that this paper uses a variety of units and performs island operation to simulate the microgrid dispatch process. At the same time, Figure 10 shows the convergence curves of multiple algorithm comparisons. Similar to the situation of grid-connected operation, MFSMA has faster convergence speed and more accurate solutions when facing mainstream swarm intelligence algorithms and can obtain smaller costs. Compared with DBO, which is closest to MFSMA, the solution accuracy of MFSMA can be improved by 8.02–12.57% at the maximum number of iterations.

## 6. Conclusions

In this paper, we improved the SMA and created a multi-strategy fusion slime mould algorithm (MFSMA), which is used to solve the economic optimal dispatching problem of the microgrid, that is, to reasonably arrange the active power output of the microgrid within 24 h a day. The total cost of the microgrid operation is at its lowest, and the pollution to the environment is reduced. For the scheduling model of this microgrid, we first add reverse learning to the algorithm to expand the search range of the population. We add a modified adaptive coefficient to ensure that the algorithm can jump out in the early stage to allocate the algorithm search mode more reasonably. The local optimal solution can speed up the convergence speed in the later stage. We also added the search operator of the followers in the salp group to the algorithm, which accelerated the convergence speed of the algorithm and improved the solution accuracy. MFSMA performs better than other swarm intelligence algorithms in benchmark function testing and specific applications. In the benchmark function test, MFSMA can almost achieve the optimal solution. In the grid-connected operation, our algorithm can be 7–29% more accurate than other algorithms, and in the island operation this value can reach 8–13%. MFSMA has fast convergence speed, high solution accuracy, strong robustness, and strong positive feedback compared with other algorithms. Our contribution is mainly to adopt two operating modes and perform bad sector analysis on multiple target units. At the same time, MFSMA has significant advantages in solving microgrid optimization problems. However, our research may be limited to one microgrid unit; large scale is always the future research direction. At the same time, our articles have less research on the uncertainty of renewable energy, and so how to get closer to this reality is one of the future directions. At the same time, in the face of the latest algorithms, MFSMA must also be constantly updated, and more novel search modes may bring better results. In future research, better search operators can be added to the algorithm, and more renewable energy can be added to the microgrid scheduling model, such as hydroelectric power generation units, to further reduce the environmental pollution of the units at the same time. Research on multi-region joint scheduling for the power supply will also make the model more complex and have higher requirements for the algorithm.

## Figures and Tables

**Figure 1 biomimetics-09-00138-f001:**
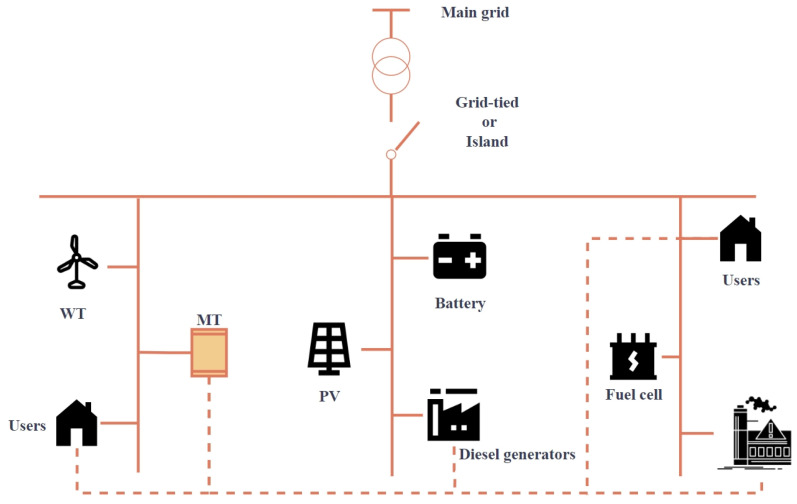
The microgrid model.

**Figure 2 biomimetics-09-00138-f002:**
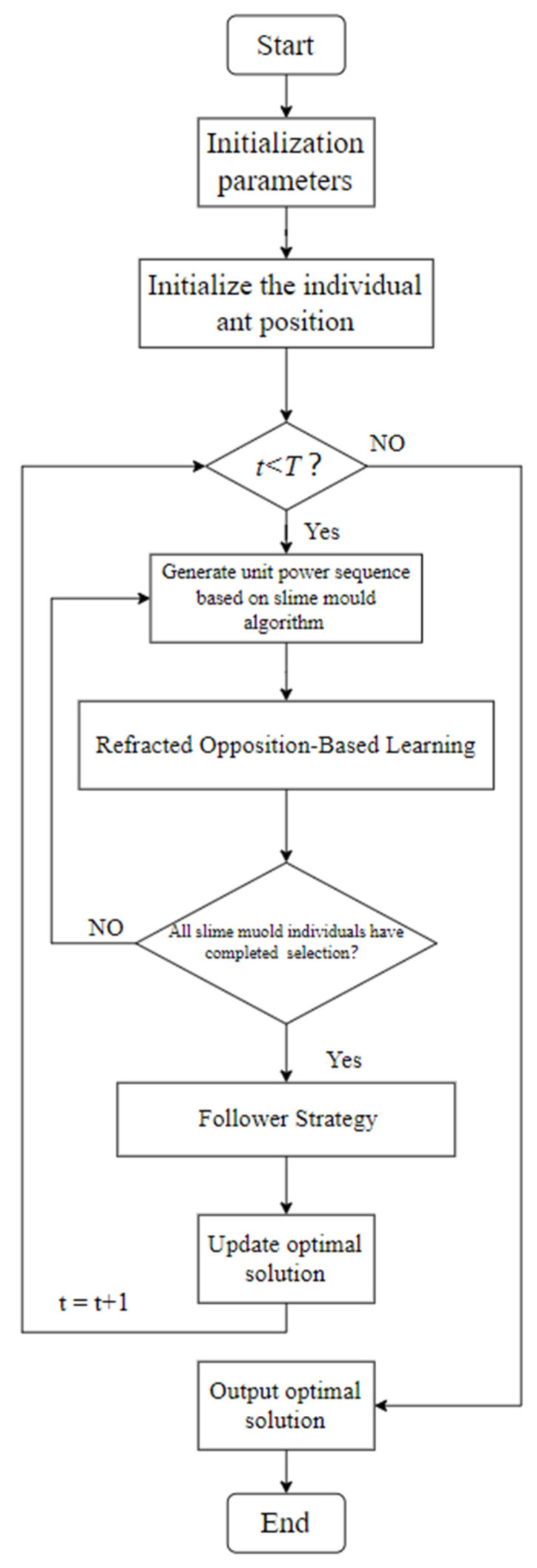
MFSMA flowchart.

**Figure 3 biomimetics-09-00138-f003:**
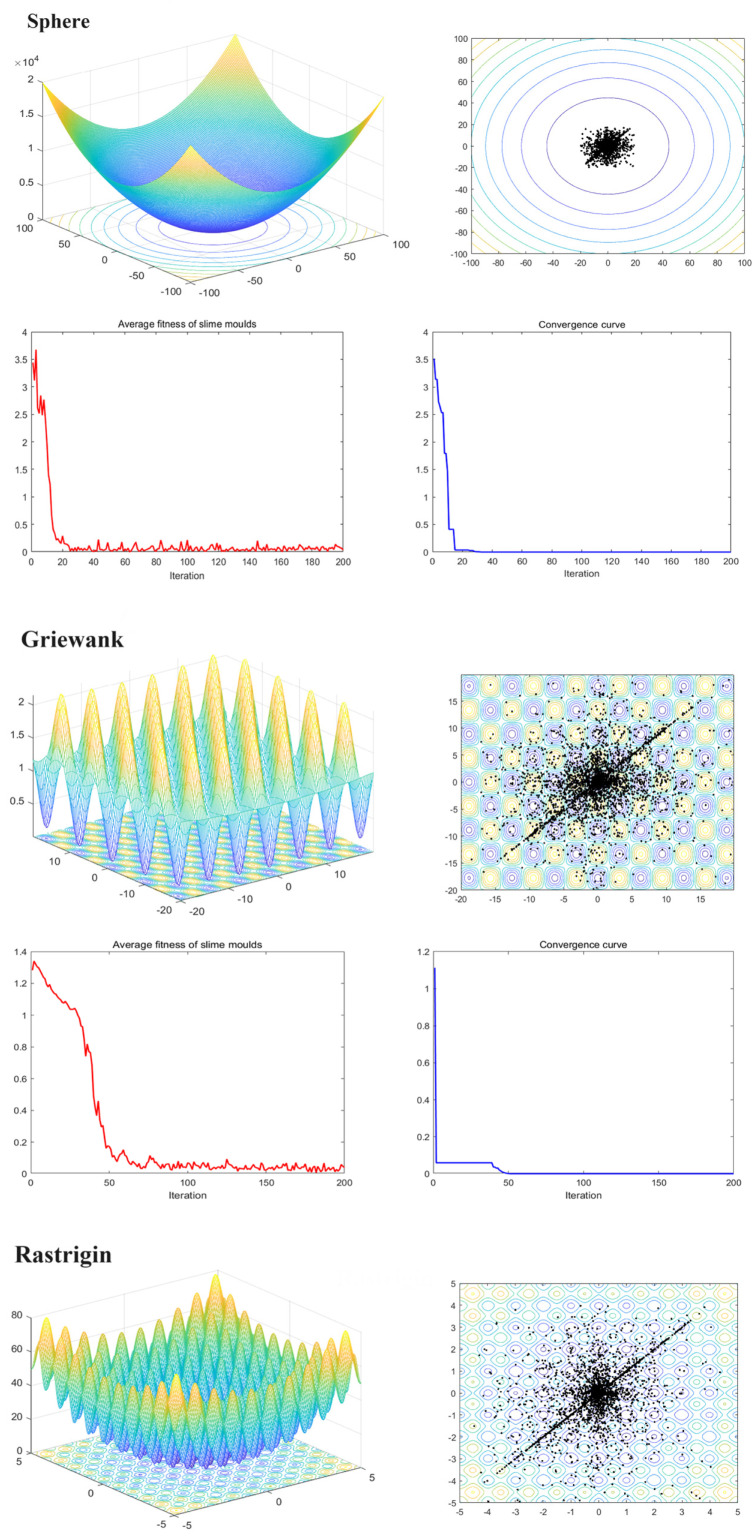
MFSMA qualitative analysis.

**Figure 4 biomimetics-09-00138-f004:**
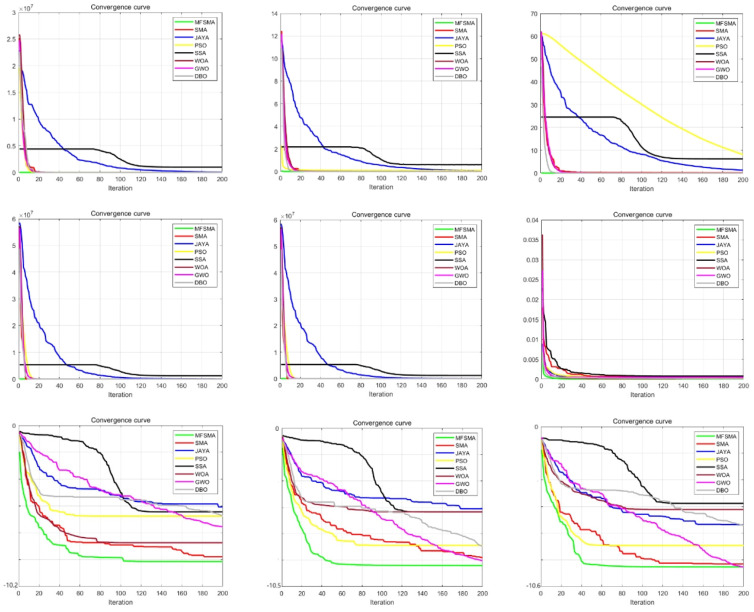
Comparison of convergence curve.

**Figure 5 biomimetics-09-00138-f005:**
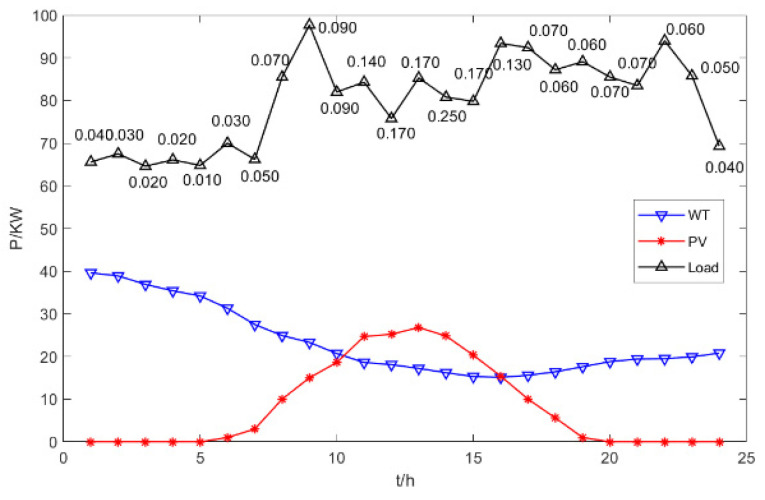
Typical 24 h loads and real-time electricity prices.

**Figure 6 biomimetics-09-00138-f006:**
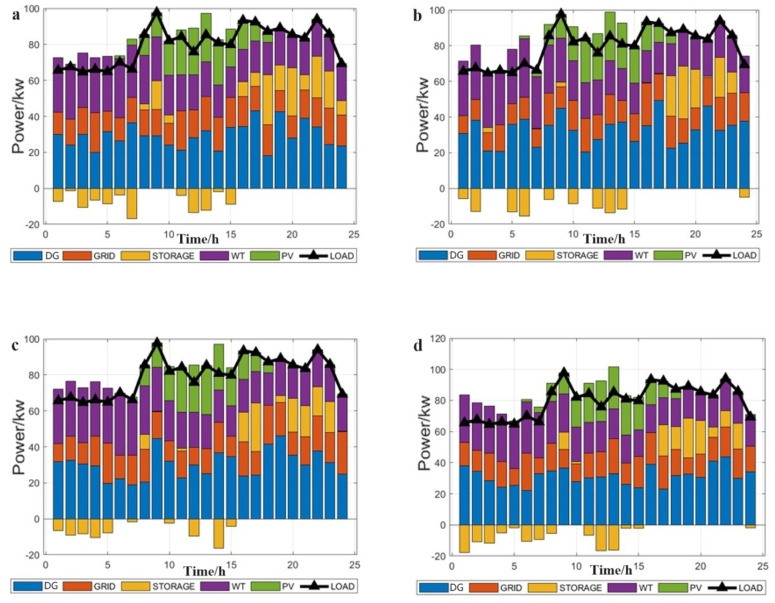
Optimize scheduling.

**Figure 7 biomimetics-09-00138-f007:**
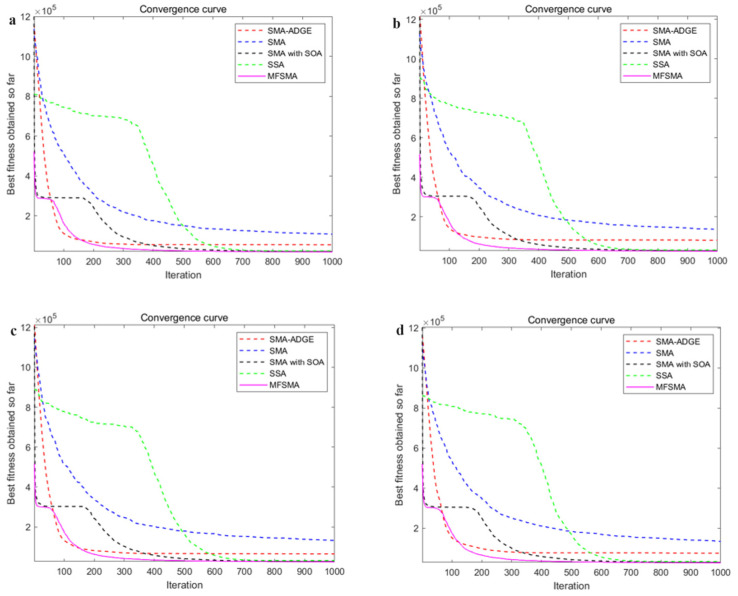
Compare the total costs obtained by different algorithms.

**Figure 8 biomimetics-09-00138-f008:**
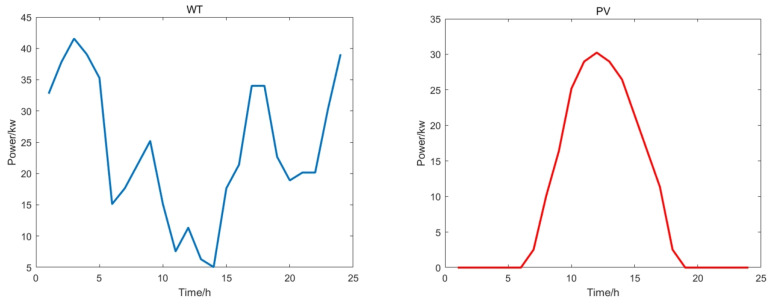
Clean energy forecast curve.

**Figure 9 biomimetics-09-00138-f009:**
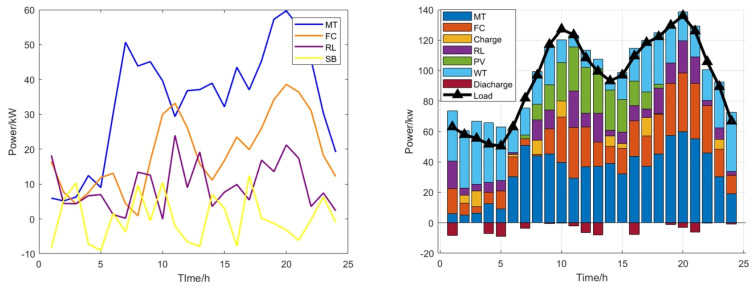
Scheduling results of island operation.

**Figure 10 biomimetics-09-00138-f010:**
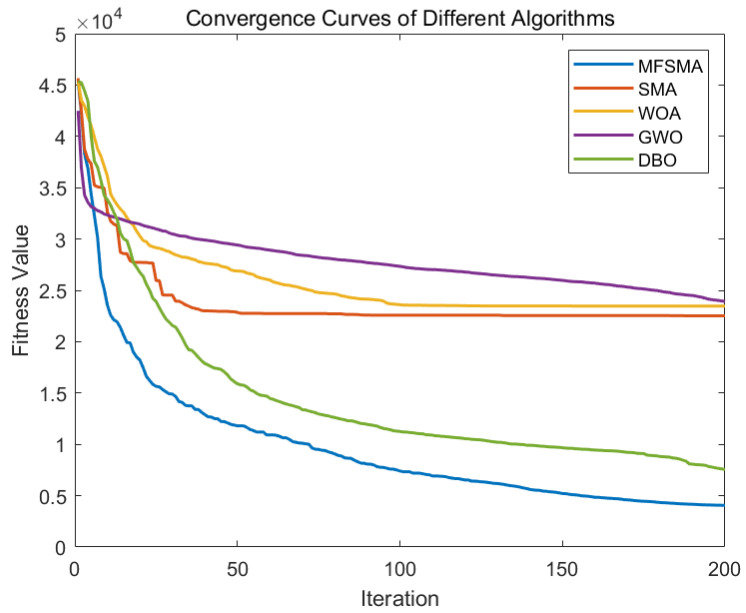
Compared curve of island operation.

**Table 1 biomimetics-09-00138-t001:** Functions.

Name	Function	Dim	Range	F_min_
F5	$\overset{n-1}{\underset{i}{\sum }}\left[100{\left(x_{i-1}-x_{i}^{2}\right)}^{2}+{\left(x_{i}-1\right)}^{2}\right]$	30	[−30, 30]	0
F7	$\overset{n}{\underset{i=1}{\sum }}ix_{i}^{4}+random\left[0,1\right)$	30	[−1.28, 1.28]	0
F11	$\frac{1}{4000}\overset{n}{\underset{i=1}{\sum }}x_{i}^{2}-\overset{n}{\underset{i=1}{\prod }}\cos\left(\frac{x_{i}}{\sqrt{i}}\right)+1$	30	[−600, 600]	0
F12	$\frac{\pi }{n}\{10\sin\left(\pi y_{1}\right)$ $+\overset{n-1}{\underset{1}{\sum }}{\left(y_{1}-1\right)}^{2}\left[1+10\sin^{2}\left(\pi y_{i+1}\right)\right]+{\left(y_{n}-1\right)}^{2}\}$ $+\overset{n}{\underset{1}{\sum }}u\left(x_{i},10,100,4\right)$ $y_{i}=1+\frac{x_{i}+1}{4}$ $u\left(x_{i},a,k,m\right)=\left\{\begin{array}{c} k{\left(x_{i}-a\right)}^{m},x_{i}>a\\ 0-a<x_{i}<a\\ k{\left(-x_{i}-a\right)}^{m},x_{i}<a \end{array}\right.$	30	[−50, 50]	0
F13	$0.1\{\sin{\left(3\pi x_{1}\right)}^{2}+\overset{n-1}{\underset{i=1}{\sum }}{\left(x_{i}-1\right)}^{2}\left[1+\sin{\left(3\pi x_{i+1}\right)}^{2}\right]+\left(x_{n}-1\right)\left[1+\sin{\left(2\pi x_{n}\right)}^{2}\right]\left.\right\}+\overset{n}{\underset{i=1}{\sum }}u\left(x_{i},5,100,4\right)$	30	[−50, 50]	0
F15	$\overset{11}{\underset{i=1}{\sum }}{\left[a_{i}-\frac{x_{i}\left(b_{i}^{2}+b_{i}x_{2}\right)}{b_{i}^{2}+b_{i}x_{3}+x_{4}}\right]}^{2}$	4	[−5, 5]	0.0003
F21	$-\overset{5}{\underset{i=1}{\sum }}{\left[\left(X-a_{i}\right){\left(X-a_{i}\right)}^{T}+c_{i}\right]}^{-1}$	4	[0, 10]	−10.2
F22	$-\overset{7}{\underset{i=1}{\sum }}{\left[\left(X-a_{i}\right){\left(X-a_{i}\right)}^{T}+c_{i}\right]}^{-1}$	4	[0, 10]	−10.4
F23	$-\overset{10}{\underset{i=1}{\sum }}{\left[\left(X-a_{i}\right){\left(X-a_{i}\right)}^{T}+c_{i}\right]}^{-1}$	4	[0, 10]	−10.5

**Table 2 biomimetics-09-00138-t002:** Specific comparison.

Function	Metric	MFSMA	SMA	JAYA	PSO	SSA	WOA	GWO	DBO
F5	AVE STD	**0** **0.028**	72.769 121.883	4.856 × 10^−5^ 2.280 × 10^−5^	9.391 × 10^−3^ 7.540 × 10^−3^	6.616 × 10^−6^ 6.280 × 10^−6^	28.640 0.180	27.638 0.872	27.025 0.5787
F7	AVE STD	**2.036 × 10^−4^** **1.728 × 10^−4^**	0.062 0.037	0.791 0.397	0.709 0.287	4.519 2.0492	0.007 0.007	0.007 0.004	0.002 0.002
F11	AVE STD	**0** **0**	0.944 0.544	12.1465 3.609	79.203 11.058	67.745 26.541	0.013 0.057	0.018 0.023	3.771 × 10^−11^ 1.686 × 10^−10^
F12	AVE STD	**4.004 × 10^−5^** **4.391 × 10^−5^**	0.069 0.132	8.541 × 10^−4^ 2.013 × 10^−5^	4.860 2.245	9.064 × 10^−6^ 1.119 × 10^−7^	0.093 0.100	0.079 0.047	0.003 0.003
F13	AVE STD	**1.981 × 10^−5^** **0.0038**	0.720 1.412	5.166 × 10^−5^ 4.656 × 10^−5^	35.670 34.638	2.698 × 10^−7^ 4.050 × 10^−7^	0.935 0.417	1.072 9.707 × 10^−4^	1.294 0.584
F15	AVE STD	**3.934 × 10^−4^** **2.023 × 10^−6^**	0.002 0.001	6.463 × 10^−4^ 2.104 × 10^−4^	0.002 0.005	0.007 0.007	0.001 8.009 × 10^−4^	0.004 0.007	8.6686 × 10^−4^ 4.529 × 10^−4^
F21	AVE STD	**−10.152** **5.605 × 10^−4^**	−9.859 0.405	−5.483 2.174	−8.769 2.509	−4.934 2.563	−7.289 2.892	−8.622 2.377	−7.219 2.743
F22	AVE STD	**−10.396** **7.164 × 10^−4^**	−9.872 0.748	−7.234 3.039	−8.278 3.367	−5.724 3.365	−6.576 2.738	−10.390 0.008	−6.662 3.276
F23	AVE STD	**−10.531** **5.655 × 10^−4^**	−10.026 0.908	−6.658 2.813	−7.743 3.908	−5.2683 3.365	−5.510 3.415	−10.523 0.009	−7.852 3.467

**Table 3 biomimetics-09-00138-t003:** Coal consumption characteristic parameters.

	$a_{i}\;\left(\$/h\right)$	$b_{i}\;\left(\$/mW\cdot h\right)$	$c_{i}\;\left(\$/mW^{2}h\right)$	$d_{i}\;\left(\$/h\right)$	$e_{i}\;\left(rad/mW\right)$	$P_{min}\;\left(kW/h\right)$	$P_{max}\;\left(kW/h\right)$
DG1	0.26	−0.3975	0.002176	0.02697	−3.975	15	45.5
DG2	−95.14	0.4846	0.00001176	−0.05914	4.864	20	50
DG3	−53.99	0.4462	0.0001498	−0.05399	4.462	19	49
DG4	−61.13	0.5084	0.0000416	−0.06113	5.084	20	49

**Table 4 biomimetics-09-00138-t004:** Microgrid pollution control costs.

Pollutant	Cost ($/kg)	WT(PV) (g/kW·h)	MT (g/kW·h)	DG (g/kW·h)	FC (g/kW·h)	Grid (g/kW·h)
$CO_{2}$	0.03	0	724	1488	489	889
$SO_{2}$	2.1468	0	0.0036	0.01388	0.003	1.8
$NO_{x}$	9.1074	0	0.2	0.3155	0.014	1.6

**Table 5 biomimetics-09-00138-t005:** Power upper and lower limits.

	WT (kW)	FC (kW)	SB (kW)	Removable Load (kW)
$P_{min}$	0	0	−20	0
$P_{max}$	65	50	20	20

## Data Availability

All data included in this study are available upon request by contact with the corresponding author.

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
