# Peer review of "Research on Microgrid Optimal Dispatching Based on a Multi-Strategy Optimization of Slime Mould Algorithm"

_biomimetics, 2024, doi:10.3390/biomimetics9030138_

Round 1

Reviewer 1 Report (New Reviewer)

Comments and Suggestions for Authors

This work presents ‘Microgrid optimal dispatching based on multi-strategy slime mould optimizer in an interesting manner. While the work is presented engagingly, I recommend considering the following points for further refinement.

1- The paper requires further English revision 

2-   The four illustrations within Figures 4 and 5 should be labeled as (a), (b), (c), and (d).

3-  Revise the insertion of references to ensure they are placed correctly within the text.

4- Verify the clarity of the figures in the text as some appear to be blurry.

Comments on the Quality of English Language

This work presents ‘Microgrid optimal dispatching based on multi-strategy slime mould optimizer in an interesting manner. While the work is presented engagingly, I recommend considering the following points for further refinement.

1- The paper requires further English revision 

2-   The four illustrations within Figures 4 and 5 should be labeled as (a), (b), (c), and (d).

3-  Revise the insertion of references to ensure they are placed correctly within the text.

4- Please verify the clarity of the figures in the text as some appear to be blurry.

Author Response

Response to Reviewer 1 Comments

1. Summary

2. Questions for General Evaluation

Reviewer’s Evaluation

Response and Revisions

Does the introduction provide sufficient background and include all relevant references?

Can be improved

We have modified the introduction

Are all the cited references relevant to the research?

Can be improved

We have updated the references

Is the research design appropriate?

Yes

Are the methods adequately described?

Yes

Are the results clearly presented?

Can be improved

We have updated the experimental results

Are the conclusions supported by the results?

Can be improved

We have updated the presentation format of the results

3. Point-by-point response to Comments and Suggestions for Authors [in red]

Comments 1: The paper requires further English revision 

Response 1: Thank you for pointing this out. We agree with this comment. Therefore, we have corrected incorrect grammar and incorrect sentences to make the article smoother

​

Comments 2: The four illustrations within Figures 4 and 5 should be labeled as (a), (b), (c), and (d).

Response 2: Agree. We have added the label. Because the article has been modified, the current serial numbers of these two pictures are Figure 5 and Figure 6.

Comments 3: Revise the insertion of references to ensure they are placed correctly within the text.

Response 3: Agree, the reference does have a lot of incorrect identification, we have corrected it.

Comments 4: Verify the clarity of the figures in the text as some appear to be blurry.

Response 4: Agreed, we've replaced a lot of the blurry drawings and updated some images.

4. Response to Comments on the Quality of English Language

Point 1: The paper requires further English revision 

Response 1: Thank you for pointing this out. We agree with this comment. Therefore, we have corrected incorrect grammar and incorrect sentences to make the article smoother

​

Point 2: The four illustrations within Figures 4 and 5 should be labeled as (a), (b), (c), and (d).

Response 2: Agree. We have added the label. Because the article has been modified, the current serial numbers of these two pictures are Figure 5 and Figure 6.

Point 3: Revise the insertion of references to ensure they are placed correctly within the text.

Response 3: Agree, the reference does have a lot of incorrect identification, we have corrected it.

Point 4: Verify the clarity of the figures in the text as some appear to be blurry.

Response 4: Agreed, we've replaced a lot of the blurry drawings and updated some images.

5. Additional clarifications

Thank you for the review. During the review time of the article, we have optimized and modified the article according to our own ideas. We hope this will be helpful to the review process of the article.

Reviewer 2 Report (New Reviewer)

Comments and Suggestions for Authors

Please refer to the attached pdf file.

Comments on the Quality of English Language

The quality of English is in general good.

Author Response

Response to Reviewer 2 Comments

1. Summary

2. Questions for General Evaluation

Reviewer’s Evaluation

Response and Revisions

Does the introduction provide sufficient background and include all relevant references?

Must be improved

We have modified the introduction

Are all the cited references relevant to the research?

Can be improved

We have updated the references

Is the research design appropriate?

Must be improved

We optimized the research design

Are the methods adequately described?

Must be improved

We added experiments and descriptions

Are the results clearly presented?

Can be improved

We have updated the experimental results

Are the conclusions supported by the results?

Can be improved

We have updated the presentation format of the results

3. Point-by-point response to Comments and Suggestions for Authors [in green]

Comments 1: Concerning the optimization model presented in Section 2, I have found the modelling approach a bit confused. In my opinion, in a revised version, it would be important to follow a more canonical and rigorous Mathematical Programming approach. First,the Authors should provide a clear concise statement of the problem, introducing all the system elements, and then proceed to describe in sequence: 1) the decision variables, 2)the feasibility constraints, 3) the objective function(s).

In the current version, I have found the definition of decision variables particularly naive.

For example, in formula (1) I would expect the quantities: 

to be continuous decision variables, as in a canonical unit commitment problem; however,

this is not well-stated.

Response 1: I agree with you that there are indeed big problems with the expression of various formulas in Section 2. We have made a lot of modifications to this part, and clarified the specific formulas of decision variables, constraints and objective functions, and corrected the errors of formulas and expressions.

​

Comments 2: In Subsection 3.3.4, the Authors present "The MFSMA for solving microgrid optimal dispatching problem". However, what is missing in the section is a detailed discussion of how the specific features of the microgrid optimization problem are mapped to the elements of the metaphor of the new slime mould algorithm.

In other words, the Authors just present the general pseudocode of the MFSMA, but it

would be important to also provide the version of the MFSMA adapted to the microgrid

problem.

Response 2: Agree. We have updated the table of pseudocode and used green text to describe the specific search process of MFSMA in the paragraphs after the pseudocode.

​

Comments 3: In the review of related literature, I think that it would be important to better discuss the specific merits and limits of each work. In particular, it would be better to avoid blocks of 1citations like [14][15][16][17] on page 2 and provide more details about each work. A better analysis of merits and limits could also allow the Authors to refine the statement of the original contributions of this manuscript.

Response 3: Agree. We have carried out a more detailed expansion of the references in [14]-[17].

Comments 4: In my opinion, the introduction and the review of related literature should be enlarged and strengthened providing additional recent references of works that have dealt with optimization methods for microgrids, also taking into account the sources of data uncertainty that naturally affect them. To this end, I think the Authors should also include and discuss the following works:

(a) "Robust opportunistic optimal energy management of a mixed microgrid under asymmetrical uncertainties" by A. Nammouchi et al., Sustainable Energy, Grids and Networks, 2023, DOI: 10.1016/j.segan.2023.101184

(b) "Optimum control of power flow management in PV, wind, and battery-integrated

hybrid microgrid systems by implementing in real-time digital simulator-based platform" by S. Patel et al., Soft Computing, 2023, DOI: 10.1007/s00500-023-07838-1

Response 4: Agree, we have added the latest papers and research results to the review, analyzed the pros and cons, and added the two references you recommended in [33] [34].

Comments 5: In my opinion, the discussion about the computational results for the unimodal and multimodal test functions is quite limited and should be enlarged, by providing additional insights.

Response 5: Agree, we expand the discussion and comparison in Section 4, adding the presentation form and comparative content of data analysis.

Comments 6: The conclusive section would profit from a larger discussion about the limits of this work, especially in terms of aspects of the new MFSMA that could still be improved. Enlarging the discussion would also allow to better identify and state possible directions for future research.

Response 6: Agree, we have included a discussion of the limitations of the article in the conclusion.

Reviewer 3 Report (New Reviewer)

Comments and Suggestions for Authors

The current work proposes a multi-strategy fusion slime mould algorithm to solve the microgrid optimal dispatching problem. The paper design needs more attention to acheive the target aims with deep discussion. Authors are welcome to consider the following issues to improve the paper quality: 

1. Update both abstract and conclusion sections design to summerize the paper contributions.

2. Survey part is limited and must extended to cover the recent application of slime mould optimizer and other frameworks of microgrid, in general what are the research gap and the actual contribution in your own work. 

3. In section 5.2, be clear with the objectives and constraints you consider in this work. The problem is cost based minimzation only. then no multiple objectives were considered, what are the limitations for lower and upper limits in Eq. 10?

4. Consider the uncertainty of the intermittent sources is needed. 

5. Add the flowchart for the considered solution methodolgy of the optimal economic operation microgrid. 

6. How do you select the parameters in Table 1 for competitive algorithms?

7. Check typo errors in tables 4 and 5. Comment on table 5 is limited. 

8. Add the indices for success of your algorithm compared to others.

9. Enhance conclusion section with numerical findings and future trend. 

Comments on the Quality of English Language

Needs more attention

Author Response

Response to Reviewer 3 Comments

1. Summary

2. Questions for General Evaluation

Reviewer’s Evaluation

Response and Revisions

Does the introduction provide sufficient background and include all relevant references?

Must be improved

We have modified the introduction

Are all the cited references relevant to the research?

Must be improved

We have updated the references

Is the research design appropriate?

Can be improved

Are the methods adequately described?

Are the results clearly presented?

Are the conclusions supported by the results?

3. Point-by-point response to Comments and Suggestions for Authors [in blue]

Comments 1: Update both abstract and conclusion sections design to summerize the paper contributions.

Response 1: Agree, we added the main contributions of this paper in the Abstract, Introduction and Conclusion.

​

Comments 2: Survey part is limited and must extended to cover the recent application of slime mould optimizer and other frameworks of microgrid, in general what are the research gap and the actual contribution in your own work. 

Response 2: Agree, the latest SMA and microgrid-related research has been added to the introduction, and their research content has been summarized. and put forward their limitations

Comments 3: In section 5.2, be clear with the objectives and constraints you consider in this work. The problem is cost based minimzation only. then no multiple objectives were considered, what are the limitations for lower and upper limits in Eq. 10?

Response 3: Agree, we have optimized all aspects of the description of the research content, re-clarified the objectives and limitations, and updated the descriptions of each formula in Section 2. The upper and lower limits in the equation will be given in the table. For example,  in Table 3 is the lower limit of the output power of the micro turbine.

Comments 4: Consider the uncertainty of the intermittent sources is needed.

Response 4: I agree with you. We attribute the study of uncertainty to future research content, because for example, the use of machine learning to predict renewable energy is also something that can be expanded in large quantities. This part is relatively large and is not suitable to be added to the simulation of microgrid. The text focuses on the optimal dispatch of microgrids. This article focuses more on scheduling known information rather than obtaining information. If you use a simple way to predict, such as wind power, it is not innovative; if you use a new method to predict data uncertainty, it may be a different kind of research content.

Comments 5: Add the flowchart for the considered solution methodolgy of the optimal economic operation microgrid.

Response 5: Agree, we added the flow chart (Figure 2).

​Comments 6: How do you select the parameters in Table 1 for competitive algorithms?

Response 6: The parameters in the table are set according to the default parameters in the author's paper. The fact is that during the subsequent revision process, we changed the competition algorithm and believed that this parameter table did not have much significance and had deleted it.

Comments 7: Check typo errors in tables 4 and 5. Comment on table 5 is limited. 

Response 7: Agreed, we did get the spelling of some units wrong, such as mW/MW, and we also added new pollution indicators for research volumes. The current serial numbers of these two tables are Table 3 and Table 4.

Comments 8: Add the indices for success of your algorithm compared to others.

Response 8: Agreed, we have added specific quantitative values ​​in the comparison, as shown in the lower part of Figure 7. At the same time, we optimized the comparison content in Table 2 and added a convergence diagram (Figure 4) to the comparison of benchmark algorithms.

Comments 9: Enhance conclusion section with numerical findings and future trend. 

Response 9: Agreed. We added numerical comparisons, the contribution of this paper, the limitations of this paper, and our future research directions in the summary.

Reviewer 4 Report (New Reviewer)

Comments and Suggestions for Authors

Dear authors, 

the paper proposes a modified slime mould algorithm called multi-stategy fusion slime mould algorithm to deal with the economical optimal dispatiching problem of a microgrid. The optimization algorithm is detailed throughout the text as well as the model of the generators used in the microgrid. Some revisions should be addressed in order to facilitate the reading of the paper: 

1) Which title is the correct one? "Research on microgrid optimal dispatching based on a mul-ti-strategy optimization of Slime Mould Algorithm" or "Microgrid optimal dispatching based on multi-strategy slime mould 0ptimizer"?

2) line 138 - which charateristics terms a, b, c, d and e are related to? Please identify them in the text. 

3) Equation (1) - insert a space before and after the equation. It eases the reading of it.

4)  line 151 - line 138 - which charateristics terms k0, k1 and k2 are related to? Please identify them in the text. 

5) line 209 - n represents the number of units - where is term n in equation 5?

6) Figure 2 - The figures are very small. The title and axis are impossible to read. 

7) figure 4 - which units are axis x and y on figure 4?

Comments on the Quality of English Language

1) line 192 - "Where P and Wmax are the charge/discharge power and discharging power" - I understood that P is the charge/discharge power. The phrase is not well written.  

2) line 252/equation 9 - Would the right term be Pgrid,t instead of Pgird,t?

3) line 288 - although mold is also correct, you have used mould throughout the text. Try to use only one form: either mould or mold. 

4) line 453 and 465 - Separate the word "table" from its number, i.e. table 4 and table 5. 

5) line 470 - "temperature. photovoltaic generators" - letter p of photovoltaic should be capitalized. . 

6) line 529 - "within 24 hours a day, The total cost" . It should be a final point after the word "day". 

Author Response

Response to Reviewer 4 Comments

1. Summary

2. Questions for General Evaluation

Reviewer’s Evaluation

Response and Revisions

Does the introduction provide sufficient background and include all relevant references?

Yes

We have modified the introduction

Are all the cited references relevant to the research?

Yes

We have updated the references

Is the research design appropriate?

Can be improved

We optimized the research design

Are the methods adequately described?

Can be improved

We added experiments and descriptions

Are the results clearly presented?

Can be improved

We have updated the experimental results

Are the conclusions supported by the results?

Yes

We have updated the presentation format of the results

3. Point-by-point response to Comments and Suggestions for Authors [in purple]

Comments 1: Which title is the correct one? "Research on microgrid optimal dispatching based on a mul-ti-strategy optimization of Slime Mould Algorithm" or "Microgrid optimal dispatching based on multi-strategy slime mould 0ptimizer"?

Response 1: In fact, we have previously modified the title of the paper based on the opinions of other reviewers, so there will be ambiguities. Now we have unified the title of the paper and are consistent with the title displayed on the website. ​

Comments 2: line 138 - which charateristics terms a, b, c, d and e are related to? Please identify them in the text. 

Response 2: Agree,  is explained again in 2.1.

Comments 3: Equation (1) - insert a space before and after the equation. It eases the reading of it.

Response 3: Agree, inserted it.

Comments 4: line 151 - line 138 - which charateristics terms k0, k1 and k2 are related to? Please identify them in the text.

Response 4: Agreed, we have updated this formula and explained this series of coefficients in the new 2.2.

Comments 5:line 209 - n represents the number of units - where is term n in equation 5?

Response 5: Agreed, this formula description is wrong. We have updated the formula and explained the formula. Currently, this content is in formula (7).

Comments 6:Figure 2 - The figures are very small. The title and axis are impossible to read. 

Response 6: In fact, this is a very high-resolution picture, and it will be very clear if you zoom in. Now we have divided it into several parts, so that it can be seen clearly at normal scale, but it will take up a lot of space. (Figure 3)

Comments 7:figure 4 - which units are axis x and y on figure 4?

Response 7: Agree, we added the x-axis and y-axis labels to Figure 6.

4. Response to Comments on the Quality of English Language

Point 1: line 192 - "Where P and Wmax are the charge/discharge power and discharging power" - I understood that P is the charge/discharge power. The phrase is not well written.   

Response 1: Agree, this place was indeed not written clearly enough at the time. However, we later revised the formula and explanation of energy storage batteries, so this part of the error no longer exists.

​

Point 2: line 252/equation 9 - Would the right term be Pgrid,t instead of Pgird,t?

Response 2: Agreed, although we modified this part of the formula, we changed all the wrong ‘gird’ to ‘grid’.

Point 3: line 288 - although mold is also correct, you have used mould throughout the text. Try to use only one form: either mould or mold.

Response 3: Agreed, we now use 'mould'.

Point 4:  line 453 and 465 - Separate the word "table" from its number, i.e. table 4 and table 5. 

Response 4: Agreed, we separated them.

Point 5: line 470 - "temperature. photovoltaic generators" - letter p of photovoltaic should be capitalized.

Response 5: Agreed, we modified it.

Point 6:  line 529 - "within 24 hours a day, The total cost. It should be a final point after the word "day". 

Response 6: Agreed, we modified it.

5. Additional clarifications

Thank you for the review. During the review time of the article, we have optimized and modified the article according to our own ideas. We hope this will be helpful to the review process of the article.

Round 2

Reviewer 2 Report (New Reviewer)

Comments and Suggestions for Authors

The Authors have addresses all my comments and the overall quality and readability of the paper have been improved.

As final (very) minor comment, associated with recommendation for acceptance, I suggest to the Authors to reduce the size of the text in Figure 3 (e.g, "Rosenbro" and "Ackley"), since it looks too big with respect to the size of the shown diagrams.

Author Response

Response to Reviewer 2 Comments

1. Summary

2. Questions for General Evaluation

Reviewer’s Evaluation

Response and Revisions

Does the introduction provide sufficient background and include all relevant references?

Yes

Are all the cited references relevant to the research?

Yes

Is the research design appropriate?

Yes

Are the methods adequately described?

Yes

Are the results clearly presented?

Yes

Are the conclusions supported by the results?

Yes

3. Point-by-point response to Comments and Suggestions for Authors [in green]

Comments 1: As final (very) minor comment, associated with recommendation for acceptance, I suggest to the Authors to reduce the size of the text in Figure 3 (e.g, "Rosenbro" and "Ackley"), since it looks too big with respect to the size of the shown diagrams. 

Response 1: Agree. We changed the font size in Figure 3.​

Reviewer 3 Report (New Reviewer)

Comments and Suggestions for Authors

No further issues 

Comments on the Quality of English Language

Fine

Author Response

1. Summary

2. Questions for General Evaluation

Reviewer’s Evaluation

Response and Revisions

Does the introduction provide sufficient background and include all relevant references?

Yes

Are all the cited references relevant to the research?

Yes

Is the research design appropriate?

Can be improved

Are the methods adequately described?

Can be improved

Are the results clearly presented?

Yes

Are the conclusions supported by the results?

Yes

This manuscript is a resubmission of an earlier submission. The following is a list of the peer review reports and author responses from that submission.

Round 1

Reviewer 1 Report

Comments and Suggestions for Authors

Comments on the Quality of English Language

Moderate editing of English language required.

Author Response

Dear reviewer,

Thank you for your letter and for the reviewers’ comments concerning our article biomimetics-2511177 entitled "Research on microgrid optimal dispatching based on a multi strategy optimization of Slime Mould Algorithm ", which we submitted to the Biomimetics. Those comments are all valuable and very helpful for revising and improving our paper, as well as the essential guiding significance to our research. We have studied the comments carefully and have made corrections which we hope to meet with approval.

In the revised manuscript, we have used RED text to indicate the modifications made based on your suggestions. By the way, your opinion is so representative that it may overlap with other reviewers, so the color mark may not be your color. I hope you can read the full text, so that you may understand our revisions better. The main corrections in the paper and the responses to the reviewer’s comments are as follows file.

Reviewer 2 Report

Comments and Suggestions for Authors

Optimal dispatch of sources in microgrids is an important task, allowing to fully utilize the benefits of local renewable sources while minimizing the operational costs. Evolutionary algorithms and optimization techniques are quite popular for this task. However, with respect to the complexity of the optimization problem and the need to solve the problem in acceptable short times (especially when implementing quasi-real-time control algorithms), further development in the field of EA is of great interest for the scientific community. From this point of view, the topic of this manuscript is interesting, important, and falls within the scope of this journal.

Formal evaluation:

The title of the manuscript is informative and provides a fair description of the topic of this work. The Abstract provides a sufficient description of the content of this manuscript and its main contributions.

The introduction provides some minimalistic description of the problematic, how it is covered in literature, and what is the contribution of this work. The list of references is adequate.

The layout of the manuscript must be improved. There are several problems with the layout, including captions directly at the end of the page, tables split between pages, etc. Some captions have inconsistent formatting (e.g. formatting of caption 2.5).

Quality of figures must be improved. Figures have low quality, are blurry, axis labels are hard to read. Especially quality of Figure 2 is very bad. It is barely readable even at a high zoom level. Plot titles are not readable even when using extreme zoom levels!

The formatting of tables 1 and 2 should be improved (basically the table layout). Moreover, in Table 2, try to reduce the number of decimal places where possible to improve the readability (in current form, it feels extremely messy).

The formatting of equations must be improved. There are some typos and inconsistencies in marking indexes (e.g. equation 2.2. - vci < v < vci – should be vci < v < vco), and not all variables used in equations are properly described in the following text. Moreover, please don’t use an asterisk as the multiplication sign. Although asterisk is commonly used in various programming languages, the correct symbol for multiplication is ∙ or ×.

I’m not an English native speaker and I don’t dare to make any strict judgments about the style and language quality. In general, the manuscript is easy to read and understand, however, there are some typos (for example, in equation 2.9 Pgird instead of Pgrid).

Content evaluation:

From the content point of view, I see several significant problems. First, it is not clear what is the real composition of the microgrid. In Figure 1, there is a wind turbine, solar generator, microturbine and a fuel cell. However, in subsequent sections 2.1-2.4, a storage device is described instead of a fuel cell. Moreover, most of subsection 2.4 talks about energy storage, however, formula 2.4 describes a cost function of a fuel cell! A total mess!

From Figure 4, it is clear that the model contained something labeled “storage cell”, so I suppose the model contained an energy storage device and not a fuel cell (in that case, the cost function for the energy storage is missing completely). However, in that case, the dispatch model is not taking into account the limited capacity of the storage (only the charging/discharging power is described with equation 2.12), and also the efficiency of the charging-discharging cycle is not considered! This is an unacceptable simplification compromising the validity of all related results.

Another obvious nonsense is the load curve presented in Fig. 3. A microgrid with a basic load level of 650 MW is not a microgrid but something like a smaller state (e.g. Latvia). Loading several hundred MW in a microgrid is absurd!

Moreover, there are basically no parameters for the individual sources (installed capacity, operational limits, etc.). Just some strange value Pmin, Pmax in MW/h (how should I understand this?) for the microturbines.

To make long story short – the whole microgrid model is a mess. There are unacceptable simplifications like neglected storage capacity, the model completely ignores installation and maintenance costs for wind a solar generation. The load profile is completely absurd. Based on such an experimental setup, it is not possible to make any assumptions about the performance of the proposed algorithm in the context of microgrid dispatch. Moreover, even in the case of a proper microgrid model, the experimental validation must incorporate a number of experiments for different load and generation scenarios. A single scenario is not enough for benchmarking any network optimization or control algorithm. It can be quite possible that an algorithm providing superb results for one particular scenario will provide less than-average performance in other scenarios.

This work must be deeply reworked.

Author Response

Dear Editors and Reviewers:

Thank you for your letter and for the reviewers’ comments concerning our article biomimetics-2511177 entitled "Research on microgrid optimal dispatching based on a multi strategy optimization of Slime Mould Algorithm ", which we submitted to the Biomimetics. Those comments are all valuable and very helpful for revising and improving our paper, as well as the essential guiding significance to our research. We have studied the comments carefully and have made corrections which we hope meet with approval.

In the revised manuscript, we have used GREEN text to indicate the modifications made based on your suggestions. By the way, your opinion is so representative that it may overlap with other reviewers, so the color mark may not be your color. I hope you can read the full text, so that you may understand our revisions better. The main corrections in the paper and the responses to the reviewer’s comments are as follow file.

Reviewer 3 Report

Comments and Suggestions for Authors

In the paper the connection of MFSMA to SSA and SMA should be better explained.
There are some errors here and there and some references are totally wrong (see e.g. [19] at line 66)

math notation should be improved, using a '*' for products is in most cases useless. In summations the index used should be clearly stated both in the equation and in the summation symbol. Eq. 2.5 is an example of this problem, there is a small t in the first summation written as starting point of the summation, but a t index also appears in the equation. The second summation is from 1 to n but over which index ?

Comments on the Quality of English Language

Some sentences are not much clear as they are now, see e.g. the one in line 96-98, or 119-120, 141 (units to the units ?) and so on.

Author Response

Dear Editors and Reviewers:

Thank you for your letter and for the reviewers’ comments concerning our article biomimetics-2511177 entitled "Research on microgrid optimal dispatching based on a multi strategy optimization of Slime Mould Algorithm ", which we submitted to the Biomimetics. Those comments are all valuable and very helpful for revising and improving our paper, as well as the essential guiding significance to our research. We have studied the comments carefully and have made corrections which we hope meet with approval.

In the revised manuscript, we have used BLUE text to indicate the modifications made based on your suggestions. By the way, your opinion is so representative that it may overlap with other reviewers, so the color mark may not be your color. I hope you can read the full text, so that you may understand our revisions better. The main corrections in the paper and the responses to the reviewer’s comments are as follows files

Round 2

Reviewer 1 Report

Comments and Suggestions for Authors

1- The following comments were not done completely: 2, 3, 5, 7, 10, 30.

2- Figure 2 is out of limits of page.

3- Where did you get your case study? If it is from other paper, you should compare your results with that in previous papers.

Author Response

Dear Editors and Reviewers:

Thank you for your letter and for the reviewers’ comments concerning our article biomimetics-2511177 entitled "Research on microgrid optimal dispatching based on a multi strategy optimization of Slime Mould Algorithm ", which we submitted to the Biomimetics. Those comments are all valuable and very helpful for revising and improving our paper, as well as the essential guiding significance to our research. We have studied the comments carefully and have made corrections which we hope meet with approval.

In the revised manuscript, we have used RED text to indicate the modifications made based on your suggestions. By the way, your opinion is so representative that it may overlap with other reviewers, so the color mark may not be your color. I hope you can read the full text, so that you may understand our revisions better. The main corrections in the paper and the responses to the reviewer’s comments are as follows:

Responds to the reviewer’s comments:

Reviewer 1:

1.The following comments were not done completely: 2, 3, 5, 7, 10, 30.

Response: Thank you for your careful observation, we did not do a good job on some issues.

For 2: Begin (slime mould algorithm) with small letters in the title, abstract, introduction, and other sections instead of (Slime Mould Algorithm). 

          Response: We found that some places or some algorithms do not use lowercase, and now they have been modified.

          For 3: Write (simulation results) instead of (experiments) through your paper because you did not conduct experiment, but case study is simulated.  

          Response: We have found a few imprecise places and modified them to "simulation". If there are still places that have not been modified, I’d appreciate if you could point them out.

          For 5: In line 17 of abstract, write (Microgrid model based on the MFSMA is established in this paper) instead of (This paper establishes a microgrid model based on the MFSMA). 

          Response: I think we fixed this in round one so we didn't change it, hope you can point it out if we didn't understand what it means to you, thanks!

          For 7: Begin Keywords with small letters and write semicolon (;) between Keywords instead of comma (,). 

          Response: We found a few more capitalized words and modified them.

          For 10: Mention other applications for (slime mould algorithm) in introduction.

          Response: We added SMA applications in round one, but we thought they were not representative enough, so we modified them and added descriptions.

          For 30: Add details for results of cost of each component in separate table instead of mentioning only the percentage reduction of cost in line 426.

          Response: Now we have table 6.

2.Figure 2 is out of limits of page.

Response: Figure 2 is the right size now, and the resolution has been increased.

  1. Where did you get your case study? If it is from other paper, you should compare your results with that in previous papers.

Response: We obtained experimental data from multiple references, and added many meaningful conditions and parameters to enrich our simulations according to the requirements of other reviewers. Due to limited resources, there is no theoretically the same as other papers. For your comments, we hope to further improve it in future work. By the way, our comparison is mainly the comparison of solutions to the same problem between different algorithms, which can be reflected in many papers. We use this method to reflect the superiority of our algorithm.

Thank you for your meticulous review. We believe this must have consumed a lot of your time. Thank you for your dedication and rigor, which made us dare not make mistakes in revising every question. We tried our best to improve the manuscript. We appreciate for Editors/Reviewers’ warm work earnestly and hope that the correction will meet with approval. Once again, thank you very much for your comments and suggestions.

Reviewer 2 Report

Comments and Suggestions for Authors

After reading the revised version of the manuscript I have to repeat my negative statement from the previous review report.  There are only cosmetic improvements, and all the most problematic points of this work are still present! From the formal point of view (noncritical issues):
1)    The authors declared that the quality of Fig. 2 was improved. However, I have to say the quality of Fig. 2 is still poor. The description of axes on the small plots in the right part of the figure are still barely readable, moreover, the blue plots are clearly incomplete! Honestly, I don’t see any reasonable improvement.
2)    Table 2 and algorithm 1 are still split between two pages in a way where only the header of the table is on one page and the rest of the table (algorithm) is on another.

From the content point of view (critical issues):
1)    The modeling of the energy storage unit is extremely simplified. The model is basically reduced to declaring power limits and calculation of costs. The model does not take into account the capacity and state of charge of the storage unit! This is obvious nonsense, basically discrediting the validity of all other results!
2)    From Fig. 4 it is clear that the amount of discharged energy is much higher than the amount of energy charged during the evaluated period. However, there is no mention of the initial conditions of the storage! One of the basic principles of scientific publication is the reproducibility of published results. However, in this case, nobody will be able to reproduce your results because the parameterization of the model is not clear (nobody knows what the maximal and minimal power of the storage unit is, what is the capacity and the initial state of charge, etc.). In case the storage unit was considered an unlimited source of energy, the case study is complete nonsense.
3)    According to the cover letter, the authors don’t see any problem in using absurd power levels in the case study (microgrid with a base load of 650 MW). But I assure you, it is a problem! Such an absurd parameterization shows you have no idea about the system you are trying to optimize (a fundamental mistake). If you think it does not matter what are the exact numbers and take it just like a numeric exercise, find some other area of interest and don’t try to apply your optimization techniques to power engineering problems. Proportions, power limits, and other “details” that are not important to you are pretty important for anybody trying to use your results for some real application. If you don’t care about realistic modeling, you can benchmark the performance of your algorithm using some abstract mathematical functions and you don’t have to bother about such unimportant details like charging/discharging efficiency, state of charge, losses etc. Power levels, energy levels, the composition of sources, load profiles, etc. are not just numbers. Moreover, physical systems are often not easily scalable in a way that what works at the level of 1000 units, works also for 1 000 000  or 1 000 000 000 units!
I agree that when modeling physical systems using numerical simulations, some simplifications are always necessary. However, in this case, the model is so oversimplified that it no longer represents the intended real system!

Author Response

Dear Editors and Reviewers:

Thank you for your letter and for the reviewers’ comments concerning our article biomimetics-2511177 entitled "Research on microgrid optimal dispatching based on a multi strategy optimization of Slime Mould Algorithm ", which we submitted to the Biomimetics. Those comments are all valuable and very helpful for revising and improving our paper, as well as the essential guiding significance to our research. We have studied the comments carefully and have made corrections which we hope meet with approval.

In the revised manuscript, we have used GREEN text to indicate the modifications made based on your suggestions. By the way, your opinion is so representative that it may overlap with other reviewers, so the color mark may not be your color. I hope you can read the full text, so that you may understand our revisions better. The main corrections in the paper and the responses to the reviewer’s comments are as follows:

Responds to the reviewer’s comments:

Reviewer 2:

1.The authors declared that the quality of Fig. 2 was improved. However, I have to say the quality of Fig. 2 is still poor. The description of axes on the small plots in the right part of the figure are still barely readable, moreover, the blue plots are clearly incomplete! Honestly, I don’t see any reasonable improvement.

Responds: We are very sorry, this is our problem, when saving the file, the picture quality was compressed, it was our mistake, now the picture is still clear when it is enlarged by 500%. Thank you for your careful observation.

  1. Table 2 and algorithm 1 are still split between two pages in a way where only the header of the table is on one page and the rest of the table (algorithm) is on another.

Responds: We have modified the form and content to ensure that the form will not be split after conversion to PDF. Thank you for your careful observation.

  1. About model, energy storage device and MV.

Responds: First of all, we apologize for the revisions and letters in round one.

We continued to refine the model. For energy storage devices, we have added many parameters such as the upper limit of power and the upper limit of stored energy. After consulting the information, we used a cost formula that is more realistic. In general, we have fully optimized the model for the energy storage device. Regarding the order of magnitude problem of this microgrid, we think your criticism is very reasonable, so the order of magnitude (kV) has been reduced. This is indeed a very serious issue. We think we did not do a good job in round one,and we are very sorry for our work in round one. It is worth mentioning that the effect of our experiment seems to be better after modifying the order of magnitude. Thank you for your persistence and seriousness, thank you!

Regardless of the final result, during the revision process, in your criticism, we learned a lot more than the article itself, and we will treat every future paper strictly, Thank you so much for your willingness to spend your precious time standing up for science and truth, We have benefited a lot.

We appreciate for Editors/Reviewers’ warm work earnestly and hope that the correction will meet with approval. Once again, thank you very much for your comments and suggestions.

Reviewer 3 Report

Comments and Suggestions for Authors

Some equations, those that have summations and indexes, should be made more clear. What are the indexes in each summation ? If it is a problem adding the indexes in the equation terms, at least explain them in the text.

Comments on the Quality of English Language

Still some problems, e.g. with punctuation, always put a space after a comma, never before.

Author Response

Thank you for your letter and for the reviewers’ comments concerning our article biomimetics-2511177 entitled "Research on microgrid optimal dispatching based on a multi strategy optimization of Slime Mould Algorithm ", which we submitted to the Biomimetics. Those comments are all valuable and very helpful for revising and improving our paper, as well as the essential guiding significance to our research. We have studied the comments carefully and have made corrections which we hope to meet with approval.

In the revised manuscript, we have used BLUE text to indicate the modifications made based on your suggestions. By the way, your opinion is so representative that it may overlap with other reviewers, so the color mark may not be your color. I hope you can read the full text, so that you may understand our revisions better. The main corrections in the paper and the responses to the reviewer’s comments are as follows:

Responds to the reviewer’s comments:

Reviewer 3:

  1. Some equations, those that have summations and indexes, should be made clearer. What are the indexes in each summation? If it is a problem adding the indexes in the equation terms, at least explain them in the text.

Response: For many formulas, especially those with summation, we have added instructions. This is indeed an important thing, thank you for your meticulousness.

  1. Still some problems, e.g., with punctuation, always put a space after a comma, never.

Response: There are indeed such problems, and we found several places and fixed them. If there is still this kind of problem in the article, id appreciate it if you can point it out.

We tried our best to improve the manuscript. These changes will not influence the content and framework of the paper. We appreciate for Editors/Reviewers’ warm work earnestly and hope that the correction will meet with approval. Once again, thank you very much for your comments and suggestions.
